# Isolation and Characterization of Nanobodies against a Zinc-Transporting P-Type ATPase

**DOI:** 10.3390/antib7040039

**Published:** 2018-11-07

**Authors:** Elena Longhin, Christina Grønberg, Qiaoxia Hu, Annette Susanne Duelli, Kasper Røjkjær Andersen, Nick Stub Laursen, Pontus Gourdon

**Affiliations:** 1Department of Biomedical Sciences, University of Copenhagen, Blegdamsvej 3B, DK-2200 Copenhagen, Denmark; elonghin@sund.ku.dk (E.L.); christina.groenberg@sund.ku.dk (C.G.); qiaoxia@sund.ku.dk (Q.H.); duelli@sund.ku.dk (A.S.D.); 2Department of Molecular Biology and Genetics, Aarhus University, Gustav Wieds Vej 10c, DK-8000 Aarhus C, Denmark; nsl@mbg.au.dk; 3Department of Experimental Medical Science, Lund University, Sölvegatan 19, SE-221 84 Lund, Sweden

**Keywords:** P-type ATPase, nanobody, llama, Zinc-transport, Zinc-transporting P-ATPase, ZntA

## Abstract

P-type ATPases form a large and ubiquitous superfamily of ion and lipid transporters that use ATP (adenosine triphosphate) to carry out their function. The IB subclass (P_IB_-ATPases) allows flux of heavy metals and are key players in metal detoxification, critical for human health, crops, and survival of pathogens. Nevertheless, P_IB_-ATPases remain poorly understood at a molecular level. In this study, nanobodies (Nbs) are selected against the zinc-transporting P_IB_-ATPase ZntA from *Shigella sonnei* (SsZntA), aiming at developing tools to assist the characterization of the structure and function of this class of transporters. We identify six different Nbs that bind detergent stabilized SsZntA. We further assess the effect of the Nbs on the catalytic function of SsZntA, and find that five nanobodies associate without affecting the function, while one nanobody significantly reduces the ATPase activity. This study paves the way for more refined mechanistical and structural studies of zinc-transporting P_IB_-ATPases.

## 1. Introduction

The protein superfamily of P-type ATPases is formed by phylogenetically related pumps that actively transport ions and lipids across biological membranes of prokaryotes and eukaryotes [1] at the expense of adenosine triphosphate (ATP). They are divided in five subfamilies (P_I_-P_V_) based on sequence similarity and transport specificity [2]. P_I_-ATPases transport cations, with the P_IB_-subclass being specific for heavy metals such copper and zinc. Noteworthy members of the other subfamilies include the calcium and sodium-potassium ATPases of P_II_ and the proton ATPase of P_III_. The focus here is on class 2 P_IB_-ATPases, P_IB-2_-ATPases, which comprises zinc-transporting P-type ATPases. These ATPases are relatively poorly characterized from a mechanistic and functional point of view, and only E2 states (metal-free) have been resolved structurally [3]. One reason is that metals such as zinc render these targets unstable, and another that there are no identified compounds that can bind specifically and exclusively to several specific states (including metal bound E1 conformations) of P_IB_-ATPases. The overall structural architecture is conserved in all P-type ATPases, with four domains [4]: The soluble domains, P (phosphorylation), N (nucleotide binding), and A (actuator), and the M domain in the transmembrane region. The P domain contains the highly conserved aspartic acid—lysine—threonine—glycine—threonine (DKTGT) motif with the catalytic aspartate that is targeted by ATP stimulated autophosphorylation. The N domain is responsible for orienting the ATP towards the P domain. The A domain comprises the conserved threonine—glycine—glutamic acid (TGE) loop, which allows for dephosphorylation of the catalytic aspartate in the P-domain and the M-domain is composed by a variable number of helices that enclose membranous ion-binding site(s) that are critical for transport. In addition, zinc transporting P_IB-2_-ATPases possess one or more soluble subfamily-specific domains known as heavy metal-binding domains (HMBDs), whose function remains unclear [5]. These domains work in a tightly coupled manner in order to achieve transport, and the reaction cycle is summarized in the so called Post-Albers scheme [6,7,8] (Figure 1).

Antibodies, or immunoglobulins, are large plasma proteins that play a fundamental role in protection against pathogens, such as microorganisms, and are used for numerous basic and applied science applications. Immunoglobulin gamma 1 (IgG1), which is the most abundant immunoglobulin, comprises four polypeptide chains: Two heavy chains, each formed by a variable domain (V_H_) and three constant domains (C_H_1, C_H_2, and C_H_3), and two light chains, composed by a variable (V_L_) and a constant (C_L_) domain. The paratope (antigen binding-site) is formed by the V_L_ and V_H_ domains and mediates the interaction with the antigen [9]. However, heavy-chain only antibodies are present in certain species [10]: They are smaller (about 75 kDa) than other antibody isotypes and are formed by two heavy chains, each containing a V_HH_, C_H_2, and C_H_3 domain. Their paratope permits antigen-recognition despite being formed by a single V_HH_ domain only, paving the way for the development of single-domain antibodies also called nanobodies. These engineered antibodies are derived from such heavy-chain only antibodies and consist of a single polypeptide chain (about 13 kDa) folding into a variable domain (V_HH_). They can be obtained by immunization of camelids (e.g., llamas) with the target antigen, followed by generation of phage display libraries and screening for antigen binding [11].

The aim of this work is to isolate nanobodies (Nbs) that selectively associate with the zinc-transporter, ZntA, from *Shigella sonnei* (SsZntA), the target employed previously for structural characterization of P_IB-2_-ATPases [3], to develop inhibitors for further structural and functional studies. We successfully raise and purify Nbs against SsZntA and perform experiments to assess binding and inhibition capacities. Notably, we identify six Nbs, which bind specifically to SsZntA, including one that exhibits an inhibitory effect on the ATPase activity.

## 2. Materials and Methods

### 2.1. SsZntA Production

In the following text we refer to the manufacturers Sigma-Aldrich with location in Schnelldorf, Germany; and to VWR with location in Søborg, Denmark, unless other is stated. The gene for ZntA from the bacterium *Shigella sonnei* (UniProtID Q3YW59) was cloned in the vector, pET22 (Merck, Novagen^®^, Darmstadt, Germany), containing an amino-terminus tag of eight histidine residues (HisTag) for downstream affinity chromatography purification and a cleavage site for TEV protease (TEVp) to allow removal of the HisTag. The construct, pET22-HisTag-SsZntA, was transformed into the *E. coli* C43(DE3) expression strain and cells were grown in Terrific-Broth medium (12 % peptone (Sigma-Aldrich), 24% yeast extract (Sigma-Aldrich), 4% glycerol (VWR), 50 mM Phosphate buffer pH 7 (VWR)) at 37 °C until OD600 reached 1. Then, protein production was induced with 1 mM isopropyl-β-D-thiogalactoside (IPTG) (Biosynth AG, Staad, Switzerland) at 18 °C for 24 h. Cells were harvested at 8000× *g* for 15 min, and resuspended at a concentration of 5 mL/g wet cells in buffer containing 50 mM Tris-HCl pH 8 (Sigma-Aldrich), 200 mM KCl (VWR), 20% *v*/*v* glycerol (VWR), 5 mM β-mercaptoethanol (BME) (VWR), 1 SIGMAFAST^TM^ protease inhibitor tablet (Sigma-Aldrich) per 6 L culture, and then stored at −20 °C. To the thawed cells, a final concentration of 1 mM MgCl_2_ (VWR), 2 μg/mL DNase I (Sigma-Aldrich) and 1 mM phenylmethanesulphonyl fluoride (PMSF) (Sigma-Aldrich) were added before lysis. The solution was passed through a Constant Systems cell disruptor (Constant Systems Limited, Daventry, UK) twice at 25 kpsi, large cell debris were spun down at 20,000× *g* for 40 min, and membranes were isolated by ultracentrifugation at 190,000× *g* for 3 h. The membrane pellet was resuspended in 20 mM Tris-HCl pH 7.5, 200 mM KCl, 20% *v*/*v* glycerol, 1 mM MgCl_2_, 5 mM BME, 1 mM PMSF at 3 mg (total protein) per mL (buffer), and solubilized in 1% *w*/*v* n-Dodecyl-β-D-maltoside (DDM) (Anatrace, Maumee, OH, USA) for 1.5 h, followed by ultracentrifugation at 190,000× *g* for 45 min to remove insolubilized material. The supernatant from 6 L culture was adjusted to 50 mM imidazole (Sigma-Aldrich) and 500 mM KCl prior to loading on a 5 mL HisTrap HP (GE Healthcare, Life Sciences, Uppsala, Sweden) equilibrated in buffer containing 20 mM Tris-HCl pH 7.5, 200 mM KCl, 20% *v*/*v* glycerol, 1 mM MgCl_2_, 0.015% *w*/*v* octaethylene glycol monododecyl ether (C_12_E_8_) (Nikko Chemicals Co., Ltd., Tokyo, Japan), 5 mM BME, using an Äkta pure chromatographic system (GE Healthcare, Life Sciences, Uppsala, Sweden), and was eluted with the same buffer with 500 mM imidazole added. Protein containing fractions were pooled and treated with TEVp to remove the HisTag while dialyzing to diminish the excess of imidazole. The cleaved sample was loaded on the HisTrap again (Reverse-affinity chromatography or R-IMAC) to separate uncleaved (HisTagged) protein and the TEVp; the flow through was collected and tested by Western-blot using a conjugated antibody against 6× HisTag (6× His mAb-HRP conjugated by Takara^®^ Bio Europe AB, Göteborg, Sweden) to assess cleavage. The cleaved sample was concentrated to 12 mg/mL and run on a 24 mL size-exclusion chromatography (SEC) column with Superose6 beads (GE Healthcare, Life Sciences, Uppsala, Sweden) equilibrated in *SEC buffer* (20 mM MOPS (3-(N-morpholino)propanesulfonic acid) (VWR) pH 6.8, 80 mM KCl, 20% *v*/*v* glycerol, 3 mM MgCl_2_, 0.03% *w*/*v* DDM or 0.015% *w*/*v* C_12_E_8_, 5 mM BME). The fractions corresponding to the main peak were collected, assessed for purity by SDS-PAGE (sodium dodecyl sulfate-polyacrylamide gel electrophoresis) (Thermo Fisher Scientific, Roskilde, Denmark), concentrated to 10 mg/mL, and stored at −80 °C.

### 2.2. LpCopA and MmCadA Production

CopA from *Legionella pneumophila* (LpCopA, UniProtID Q5ZWR1) and CadA from *Mesorhizobium metallidurans* (MmCadA, UniProtID I4IY19) were produced with the same buffers as SsZntA, but with somewhat different approaches. LpCopA was cloned in the pET22 vector, without any affinity chromatography tag nor cleavage site, and was purified by Ni^2+^-affinity chromatography exploiting the endogenous histidine rich amino-terminus (no engineered HisTag). MmCadA was cloned in the pET52 vector (Merck, Novagen^®^, Darmstadt, Germany) that includes an N-terminal Strep-tag II with a HRV 3C cleavage site and was purified by StrepTactin^®^ Superflow^®^ (IBA GmbH, Göttingen, Germany) [12] affinity chromatography at pH 7.8, followed by SEC at pH 7.4; this tag does not bind metal ions and therefore does not need to be removed.

### 2.3. Llama Immunization and Nanobodies Identification

Llama immunization and library generation was performed as previously described, now using a mixture of proteins including purified SsZntA for immunization [13]. Briefly, SsZntA solubilized in 0.03% *w*/*v* DDM were injected four times (100 µg/injection) during a period of 12 weeks. The immunization was performed under the permit of Capralogics Inc., which provides a healthy housing environment for all animals and adheres strictly to the United States Department of Agriculture Animal Welfare Act regulations for Animal Care and Use. Peripheral blood mononuclear cells (PMBCs) were isolated with Ficoll paque plus (GE healthcare, Life Sciences, Uppsala, Sweden), and total RNA were extracted using a RNeasy plus kit (Qiagen, Hilden, Germany). cDNA was generated with Superscript III first strand (Invitrogen) and amplified using primers specific for the VHH genes. PCR products were cloned into a phagemid vector designed to express Nbs as pIII fusions and with a C-terminal E-detection tag. VCSM13 helper phages were used for generation of a M13 phage-display library. For selection, 20 µg biotinylated SsZntA (solubilized in 0.015% *w*/*v* C_12_E_8_) bound to streptavidin beads were blocked in *SEC buffer* (containing 0.015% *w*/*v* C_12_E_8_ and supplemented with 2% *w*/*v* bovine serum albumin (BSA) (Sigma-Aldrich) for 30 min. 5 × 10^13^ M13 phage particles were incubated with the protein for 1 h before the beads were washed 15 times with *SEC buffer* (with 0.015% *w*/*v* C_12_E_8_). Elution of the bound phage particles were achieved by addition of 500 µL of 0.2 M glycine (Sigma-Aldrich) pH 2.2 for 10 min, which were added to 75 µL of 1 M Tris pH 9.1 for neutralization before being added to *E. coli* ER2738 cells. Cells were incubated for 1 h at 37˚ C and plated on agar plates with 2% *w*/*v* glucose (Sigma-Aldrich). The enriched library was amplified and used in a second round of phage display performed as the first round, but with 1 µg SsZntA and 2 × 10^12^ M13 phage particles. For ELISA, single colonies were transferred to a 96-well plate format and grown for 4 h in LB medium, before Nb production was induced by addition of IPTG to 0.8 mM. The plate was incubated by shaking overnight at 30 °C. Next, the plate was centrifuged and 50 µL of the supernatant was transferred to an ELISA plate coated with a total of 50 µg SsZntA in *SEC buffer* (with 0.015% *w*/*v* C_12_E_8_) blocked with 2% *w*/*v* BSA. After incubation for 1 h, the plate was washed four times with *SEC buffer* (with 0.015% *w*/*v* C_12_E_8_, without BME), and the anti-E-tag-HPR antibody (Bethyl Laboratories Inc., Montgomery, TX, USA) was added. The plate was then washed four times in *SEC buffer* (with 0.015% *w*/*v* C_12_E_8_, without BME) followed by the addition of 50 µL 3,3′,5,5′-tetramethyl-benzidine (Sigma-Aldrich). The reaction was quenched by addition of 50 µL of 1 M HCl and absorbance was measured at 450 nm. Positive phagemids were sequenced and subcloned in pET22, containing a PelB signal at the amino-terminus for periplasmic secretion and a 6xHisTag at the carboxyl-terminus.

### 2.4. Nanobodies Sequence Analysis

ELISA-identified Nb hits were sequenced and the obtained sequences were aligned using the SeaView software (version 4.7, PRABI-Doua, Lyon, France) [14], and a phylogenetic tree was built using the BLOSUM62 matrix. The aligned sequences were visualized using the Sequence Manipulation Suite© (Multiple Align Show) (version 2, Paul Stothard, University of Alberta, Edmonton, NA, Canada) [15] to highlight the level of identity with a threshold of 90%, which allowed selection of non-redundant Nbs for downstream efforts. The predicted secondary structure was obtained combining the predictions from ABody Builder Antibody Modelling software (Oxford Protein Informatics Group, Oxford, UK) [16] and iCAN analysis platform for nanobodies (Southeast University, Nanjing, Jiangsu Province, China) [17].

### 2.5. Nanobodies Production

The vectors containing 11 selected Nbs (Nb1 to Nb11) were transformed in *E. coli* BL21 strain and grown in 2 L Terrific Broth medium each, at 37 °C until OD600 of 0.6. The expression was induced with 1 mM IPTG at 18 °C over-night. Cells were harvested at 8000× *g* for 15 min and solubilized in 30 mL *Nb buffer* (20 mM Tris-HCl pH 7.5, 400 mM KCl, 20 mM imidazole) prior to sonication and centrifugation at 25,000× *g* for 20min. Nb proteins were purified from the clarified supernatant by incubation with 1 mL Ni^2+^ beads (Ni Sepharose^TM^ 6 Fast Flow) (GE Healthcare, Life Sciences, Uppsala, Sweden) for 1 h at 4 °C, followed by 3 washes in *Nb buffer* and elution in 15 mL *Nb buffer* with 400 mM imidazole added. Next, the eluted Nbs were concentrated to 10–15 mg/mL and run on a SEC column with *SEC buffer* (20 mM MOPS pH 6.8, 80 mM KCl, 20% *v*/*v* glycerol, 3 mM MgCl_2_, 0.015% *w*/*v* C_12_E_8_) without reducing agent. The main peak was collected, concentrated to 5 mg/mL (370 µM), and stored at −80 °C.

### 2.6. Functional Assay

#### 2.6.1. SEC Co-Elution

Samples of SsZntA (40 µM) were incubated with Nb from 1 to 11 (40 µM) in 40 mM MOPS pH 6.8, 80 mM KCl, 20% *v*/*v* glycerol, 1 mM MgCl_2_, 1 mM TCEP (tris(2-carboxyethyl) phosphine) (Sigma-Aldrich), 0.15 mg/mL C_12_E_8_ for 30 min at room temperature in a final volume of 50 µL and run on analytic-scale SEC (column volume of 2.4 mL, flow 0.04 mL/min) equilibrated with the same buffer, to separate SsZntA-Nb complex from unbound proteins. Then, fractions were run on a denaturing SDS-PAGE to visualize the presence of SsZntA and Nb. For the Nbs that showed complex formation, the assay was scaled-up to obtain more reproducible and stable signals: 40 µM SsZntA was incubated with 100 µM Nb in 50 µL and then diluted to 500 µL prior to running in a medium-scale SEC (column volume of 24 mL, flow 0.05 mL/min).

#### 2.6.2. Ni-NTA Co-Elution

In this assay, 13 nmol of SsZntA were incubated with 44 nmol of one of Nb1-2-4-5-8-9 in 20 mM MOPS pH 7.5, 200 mM KCl, 20% *v*/*v* glycerol, 1 mM MgCl_2_, 0.015% *w*/*v* C_12_E_8_, 5 mM BME, for 1 h on ice in a final volume of 500 µL. The samples were then adjusted to 400 mM KCl and 50 mM imidazole, and run on a 1 mL HisTrap HP (GE Healthcare, Life Sciences, Uppsala, Sweden) equilibrated with the same buffer at 0.3 mL/min. Fractions were collected and run on a SDS-PAGE gel, then stained with InVision™ HisTag In-gel Stain (Thermo Fisher Scientific, Roskilde, Denmark), according to the manual, to assess the presence of HisTagged proteins followed by Coomassie-staining for total protein detection.

#### 2.6.3. ATPase Activity Assay

The Baginski assay was used to measure ATPases activity and was conducted as previously described [3]. Briefly, 0.8 μM SsZntA was incubated with no Nb, 1.6 µM Nb 1 to 9, 1.6 µM AlF_3_, 2 mM AlF_3_, and 0.2, 0.4, 0.8, 1.2, 1.6, 2, 2.4, 2.8, 3.2, 6.4 µM Nb9 in 40 mM MOPS pH 6.8, 150 mM NaCl, 5 mM MgCl_2_, 20 mM (NH_4_)_2_SO_4_ (VWR), 20 mM L-cysteine (VWR), 5 mM NaN_3_ (Sigma-Aldrich), 0.25 mM Na_2_MoO_4_ (Sigma-Aldrich), 1.2 mg/mL soybean lipids (Sigma-Aldrich), 0.3% *w*/*v* C_12_E_8_, 0.5 mM ZnSO_4_ (Alfa Aesar by Thermo Fisher Scientific, Karlsruhe, Germany), in a total volume of 50 µL for 1 h at room temperature and the assay was started by addition of 5 mM ATP. The reaction was stopped after 15 min and after 30 min by adding 50 µL Stop solution (2.5% *w*/*v* ascorbic acid (Sigma-Aldrich), 0.4 M HCl, 0.48% *w*/*v* (NH_4_)_6_Mo_7_O_24_ (Fluka Analytical, Bucharest, Romania), 0.8% *w*/*v* SDS (Sigma-Aldrich)) and then 75 µL of arsenic solution (2% *w*/*v* arsenite (Sigma-Aldrich, 2% *v*/*v* acetic acid (VWR), 3.5% *w*/*v* sodium citrate (VWR)), respectively. The absorbance was measured in a microplate reader at 860 nm and the signal normalized with a sample without protein as phosphate background. When testing Nb9 with LpCopA and MmCadA, the same conditions were applied, except the use of CuSO_4_ as a metal ion with LpCopA.

## 3. Results

### 3.1. Isolation of Native SsZntA

SsZntA was produced in *E. coli* C43 cells and initially purified by immobilized metal affinity chromatography (IMAC). The 8× histidine tag (HisTag) was removed using the TEVp cleavage site, which is important since the HisTag can bind zinc and other metals, potentially interfering with activity and binding assays. Detergent solubilized SsZntA was purified by HisTag-based affinity chromatography and treated with TEVp to remove the HisTag and subjected to affinity purification again, achieving complete separation of cleaved SsZntA, uncleaved (HisTagged) SsZntA, and HisTagged TEVp (Figure 2). The flow-through of the second affinity chromatography (R-IMAC) contained completely cleaved and pure SsZntA, while uncleaved SsZntA and TEVp were present in fractions containing 250 mM and 500 mM imidazole, respectively. SsZntA was further purified using size-exclusion chromatography (SEC), achieving a high degree of purity, as assessed by SDS-PAGE, and homogeneity, as assessed by SEC (Figure 3). The final yield of SsZntA is 5 to 10 mg per liter of culture.

### 3.2. Isolation of Nanobodies

The purified SsZntA sample (in 0.03% *w*/*v* DDM) was used for llama immunization with multiple injections during a period of 12 weeks. We generated an Nb-library and, following two rounds of phage-display followed by enzyme-linked immunosorbent assay ELISA analysis, we found about 100 positive clones. These were reduced for highly redundant ones, resulting in 45 different Nb sequences that were grouped in three main families (Figure 4a). We decided to investigate 11 Nbs covering most of the overall sequence variability: Nb1, Nb2, and Nb3 belong to the first family; Nb4 and Nb5 to the second family; Nb6 and Nb7 to the third family; Nb8, Nb9, Nb10, and Nb11 did not fit in the previous families and are therefore considered outliers. Figure 4b displays the primary structures of the selected 11 Nbs, where the portions corresponding to the three complementarity-determining regions (CDRs), as well as the predicted conserved secondary structure, are indicated. We then expressed and purified the selected 11 Nbs (with HisTag) and tested them for SsZntA binding and inhibition. Expression in *E. coli* BL21 yielded 4–5 g wet cell weight, which resulted in 9–13 mg affinity-purified Nb, per liter of culture. Nb3 precipitated following affinity-chromatography and further work with it was, therefore, discontinued. The purified Nbs were subjected to SEC, achieving relatively pure Nb samples as analyzed using SDS-PAGE (Figure 5c). Purified SsZntA was tested with three different assays to assess binding (size-exclusion and nickel-affinity chromatography) and inhibitory (ATPase activity) properties of the purified Nbs.

### 3.3. Size-Exclusion Chromatography (SEC) Co-Elution

Nbs were incubated with SsZntA and subjected to SEC to assess complex formation. We tested the binding in a small-scale size-exclusion column (2.4 mL) and in a medium-scale column (24 mL). The latter was used only on Nbs, which proved to bind to SsZntA in order to obtain a more stable signal and improve the signal-to-noise ratio. The chromatogram reported in Figure 6a is acquired from the medium scale column obtained from Nb1 and is representative of those obtained from Nb2-4-5-8-9 (results not shown), and shows the first peak at 0.6 CV (14.5 mL) containing the SsZntA-Nb complex and the second peak at 0.85 CV (20.5 mL) containing unbound Nb. The overlaying dashed and dotted lines represent the chromatograms obtained from Nb1 and SsZntA, respectively, under the same conditions. As summarized in Table 1, six Nbs (Nb1-2-4-5-8-9) displayed co-elution, suggesting SsZntA-Nb interaction. Nb6-7-10-11 caused precipitation of SsZntA and hence these Nbs were not tested in subsequent assays.

### 3.4. Affinity Purification Co-Elution

The engineered HisTag of the Nbs was used to assess binding, as illustrated in Figure 7. Nb1-2-4-5-8-9 were incubated with HisTag-free SsZntA and subjected to HisTag-based affinity chromatography. The samples were then subjected to SDS-PAGE and stained with InVision^®^ HisTag In-gel staining to evaluate the presence of the HisTag in the protein bands. The results in Figure 8 represent control experiments. Panel (**a**) confirms that the HisTag-free SsZntA appears in the flow-through and is therefore lacking Ni^2+^-binding capacity; panel (**b**) shows that HisTagged SsZntA (lane +) is stained by the InVision^®^ dye, resulting in a clear band. Figure 9a reveals that binding of Nb1 to SsZntA allows for retention of the latter in the Ni^2+^-immobilized resin, proving that an Nb-SsZntA complex forms; Nb2-4-5-8-9 behaved similarly to Nb1 (results not shown). In Figure 9b, it is verified that the bands corresponding to SsZntA do not present a HisTag, while the Nbs do.

### 3.5. Nb Effect on SsZntA Functionality

We used the Baginski assay to access Nb-induced inhibition of SsZntA, which estimates the amount of released inorganic phosphate (P_i_) associated with the catalytical cycle (ATP turn-over, 1 mole P_i_ equals 1 mole of transported zinc). Different batches of Nbs were tested for inhibitory activity at a molar ratio of 1 (SsZntA) to 2 (Nb), corresponding to 8 µM SsZntA and 16 µM Nbs, yielding the results shown in Figure 10a: Wild-type SsZntA without added Nb displays a specific activity of 885 ± 87 nmol P_i_ mg^−1^ min^−1^, which is comparable to the activity previously reported [18]. Nb9 reduces the ATPase activity to around 50% of wild-type, showing a significant inhibition with a *p* value < 0.0001 (Dunn’s multiple comparison test). The inhibition on the ATPase activity by the other Nbs is non-significant at the concentrations tested. We included the phosphate analog, aluminum fluoride (AlF_3_), as a control of the ATPase inhibition assay at two different concentrations: 1.6 µM (as the Nbs) and 2 mM (previously reported [3] working concentration). AlF_3_ does not affect ATPase activity at 1.6 µM, showing that the affinity for SsZntA of Nb9 is higher than that of AlF_3_. In Figure 10b, a titration experiment of Nb9 with different molar ratios is reported: Nb9 inhibits ATPase activity of SsZntA when present at least at a molar ratio 1:1 (SsZntA:Nb9) and has the maximum inhibitory effect (at about 50%) at the molar ratio of 1:3 (SsZntA:Nb9). To assess Nb9 specificity, it was further tested against CopA from *Legionella pneumophila* (LpCopA): A member of the same subfamily as SsZntA (P_IB_-type, 34% identity to SsZntA), but transporting Cu^+^ instead of Zn^2+^, and against CadA from *Mesorhizobium metallidurans* (MmCadA); a Cd^2+^ and Zn^2+^-transporting ATPase from the same subclass as SsZntA (P_IB-2_-type, 53% identity). LpCopA and MmCadA were purified similarly to SsZntA and yield highly pure protein (Figure 11a). As shown in Figure 11b, Nb9 did not inhibit ATPase activity for any of the two proteins, suggesting that Nb9 is specific for SsZntA.

## 4. Discussion

P-type ATPases are characterized by a transmembrane domain, embedding the ion binding site(s), and a large soluble portion, which includes catalytic and regulatory domains. Each domain has a particular function that requires an overall maintained structural integrity: The relative arrangements of these domains, and their ability to undergo conformational changes, is of crucial importance for the ATPase function and the ion translocation activity [19]. Hence, ATP and phosphate analogs (e.g., AMPPCP (Adenylylmethylenediphosphonate disodium salt) [20] or AlF_3_ [21]) that occupy the nucleotide binding or phosphorylation site in the soluble portion, preventing hydrolysis, inhibit also translocation of transported ion(s) through the transmembrane domain. In this study, Nbs were selected as tool compounds, exploiting their ability to bind to cavities and active sites of proteins due to a combination of the small size and convex paratope [22]. We identified Nbs that bind specifically to the model zinc-transporting P_IB_-type ATPase SsZntA to widen the toolset available for structural and functional studies of this subclass of ATPases.

Isolated membranes containing SsZntA were solubilized in DDM, followed by detergent exchange to C_12_E_8_ using affinity chromatography. For the samples exploited for immunization, but not those used for selection, an additional detergent exchange to DDM was performed during SEC. DDM is frequently the detergent of choice for membrane proteins, providing efficient solubilization, stabilization, and preventing non-specific interaction for many membrane protein targets. This means that DDM is suitable if the aim is to reduce interaction and aggregation with other soluble and membrane proteins, as in the case of llama immunization. The SsZntA samples used for Nb selection were instead purified in 0.015% *w*/*v* C_12_E_8_: This detergent provides a smaller micelle size and, therefore, leaves more of the protein exposed for interactions. Detergents that give a small micelle size are preferred for structural biology studies, increasing the possibility for protein-protein interactions, which are essential for crystal-lattice formation [23]. In this work, the final aim was to provide new tools for SsZntA crystallization, and thus the identified Nbs need to bind and/or inhibit in the presence of the detergent used for crystallization studies of many P-type ATPases: C_12_E_8_ [3,24,25,26]. Therefore, the selection was performed using C_12_E_8_ solubilized SsZntA, but we anticipate similar results in the presence of other mild detergents, including DDM.

Following screening of the initial Nb hits, we selected 11 for further studies, attempting to address variability in the sequences. As shown in Figure 4 (**b**), the selected Nbs display distinct differences in the complementarity determining regions (CDR), while the conserved motifs and predicted scaffold structure are maintained. This observation is in line with previous findings on Nbs, since the CDRs are constituted by residues responsible for antigen binding, and different residue combinations potentially bind to different antigens. Among the selected 11 Nbs, one (Nb3) could not be purified and four (Nb6, Nb7, Nb10, Nb11) caused precipitation of the sample when mixed with SsZntA, and could thus not be tested in further assays. We believe that the observed precipitation can relate to the presence of detergent, perhaps in combination with Nb binding, which apparently compromises SsZntA further in this environment (e.g., through interaction with unstable intermediates or at sensitive regions of the proteins, such as the membrane interface). We successfully identified six Nbs (Nb1, Nb2, Nb4, Nb5, Nb8, Nb9) which bind to SsZntA, while only one (Nb9) displays significant (<50%) inhibition of the ATPase activity. Nb9 was tested against two other members of the heavy metal-transporting subfamily in order to assess its specificity: It does not inhibit the ATPase activity of neither a copper- nor a zinc-transporting ATPase, denoting a high degree of specificity and a tight interaction with specific residues and/or conformations of SsZntA, which make it an ideal tool for studying in detail the transport cycle of this protein.

Further studies are needed to assess whether Nb9 inhibits the activity by binding between domains (as allosteric inhibitors), or if it occupies the ATP or the metal binding site, thus inhibiting the activity as competitive ligands, and if it blocks the reaction cycle by binding to a specific reaction cycle state. The latter is of particular interest, since it could be the first E1-specific inhibitor to be found for a metal-transporting P-type ATPase and would thus have extensive applications in both structural and functional characterization efforts. The fact that five out of six Nbs bind, but do not inhibit, the activity may hint that these Nbs associate to, for example, the more exposed portions of the soluble domains, and not to domain-domain interfaces. In any case, they can still be exploited as tools for structural biology studies: Co-crystallization of the Nbs with SsZntA is likely to alter crystal contacts/crystal packing by, for instance, generating a wider hydrophilic surface for lattice contacts. If stronger crystal contacts are obtained, it may result in an increased resolution of the diffraction data generated from these crystals. Moreover, the identified Nbs may serve as chaperones for cryo-EM experiments, with the aim of facilitating images’ orientation, domain identification, and to render particles larger. In conclusion, this work provides the foundation for further usage of Nbs in structural studies of P_IB_-type ATPases, aiming at expanding the available toolbox to achieve a greater understanding of this important, yet elusive, subfamily of enzymes.

## Figures and Tables

**Figure 1 antibodies-07-00039-f001:**
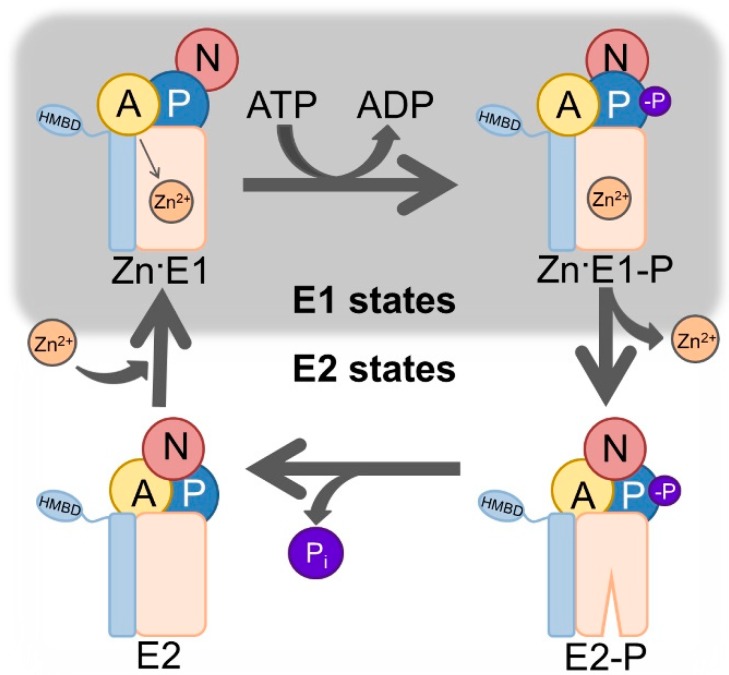
Post-Albers scheme of P_IB-2_-ATPases. The E1 (high zinc affinity) and E2 (low zinc affinity) states of the enzyme alternate, and couple ATP (adenosine triphosphate) hydrolysis to the export of zinc. The E1 state accepts one zinc (Zn^2+^) ion and ATP from the intracellular side, which promotes autophosphorylation, reaching the zinc occluded Zn·E1-P state and releasing ADP (adenosine diphosphate). Completion of phosphorylation triggers considerable conformational changes that opens the pump towards the outside, allowing release of zinc in the E2-P state. Metal discharge is associated with auto dephosphorylation, liberation of inorganic phosphate (P_i_), and allows the enzyme to reach the E2 conformation. The domains are represented as follows: The actuator (A) domain in yellow, the phosphorylation (P) domain in blue, the nucleotide-binding (N) domain in red, the transmembrane domain in light orange. Features specific for P_IB_-ATPases are shown in light blue, and includes two transmembrane helices and heavy-metal binding domain(s) (HMBD).

**Figure 2 antibodies-07-00039-f002:**
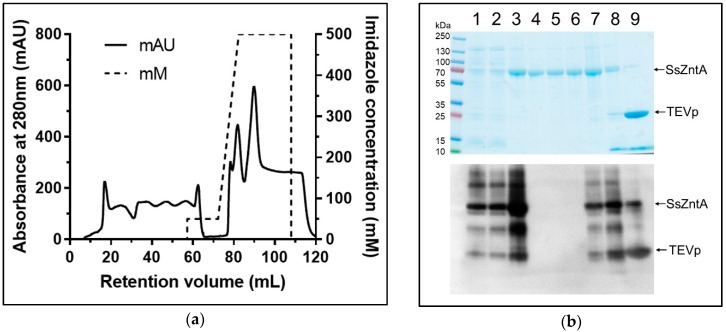
Purificaiton of the zinc-transporter SsZntA from *Shigella sonnei*: (Reverse-affinity chromatography) R-IMAC. (**a**) Reverse (second) affinity chromatography profile: The first 60 mL represent the flow-through containing cleaved SsZntA (without HisTag); at 80 mL, corresponding to 250 mM imidazole, uncleaved SsZntA is eluted; while at 90 mL, at 500 mM imidazole, HisTagged TEVp is eluted. (**b**) On top, Coomassie stained sodium dodecyl sulfate-polyacrylamide gel electrophoresis (SDS-PAGE), and on the bottom, Western blot anti-HisTag; lane 1: Solubilized membranes, lane 2: Clarified solubilized membranes, lane 3: Affinity-chromatography purified SsZntA (with HisTag), lanes 4 to 6: Flow through of reverse affinity chromatography (SsZntA without HisTag) corresponding to retention volume from 0 mL to 60 mL, lanes 7 to 9: Eluted fractions corresponding to retention volume of 80, 85, and 90 mL, respectively.

**Figure 3 antibodies-07-00039-f003:**
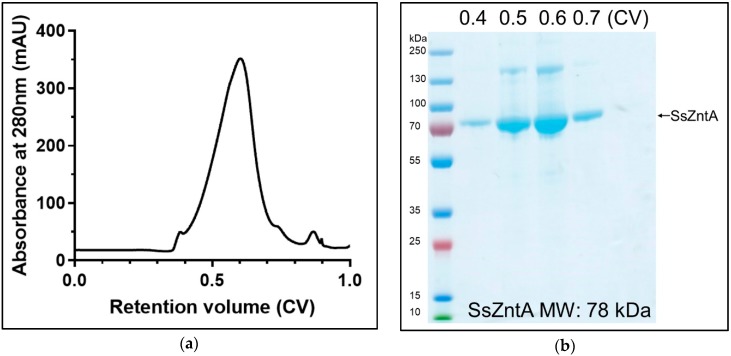
SsZntA purification: SEC. (**a**) Elution profile from a Superose6 size-exclusion column with a column volume (CV) = 24 mL: The main peak elutes at 0.6 CV, followed by minor contaminants. (**b**) Coomassie-stained SDS-PAGE containing the SEC fractions collected at 0.4, 0.5, 0.6, 0.7 CV; upper band at 150 kDa is possibly SsZntA dimers (due to SDS artifacts).

**Figure 4 antibodies-07-00039-f004:**
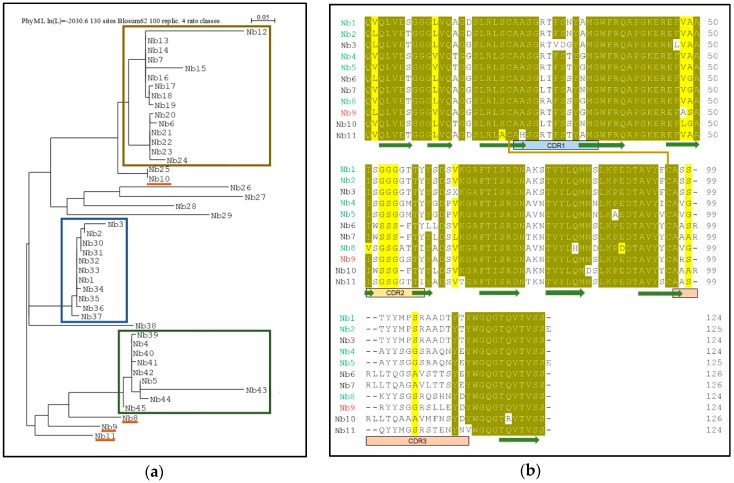
Sequence analysis of the identified Nbs. (**a**) Phylogenetic tree of 45 Nb sequences: The three main families of Nbs are highlighted by colored boxes: The first family (blue box) includes Nb1-2-3, the second family (green) includes Nb4-5, the third family (ochre) is composed of two subfamilies, which include Nb6 and Nb7, while Nb8-9-10-11 were sampled as outliers. (**b**) Sequence alignment of the 11 selected Nbs containing the predicted secondary structure (β-sheets are shown as green arrows), the complementarity-determining regions (CDR1 to 3), and the conserved disulphide bond (orange line). Identical (90% threshold) residues are in dark yellow while similar amino acids in light yellow. Nbs that bind to SsZntA, but do not inhibit activity, are represented with a green name (Nb1-2-4-5-8), while in red, the Nb that significantly inhibits ATPase activity (Nb9). The Nbs that precipitated during purification (Nb3) or that caused precipitation of SsZntA (Nb6-7-10-11) are represented in black.

**Figure 5 antibodies-07-00039-f005:**
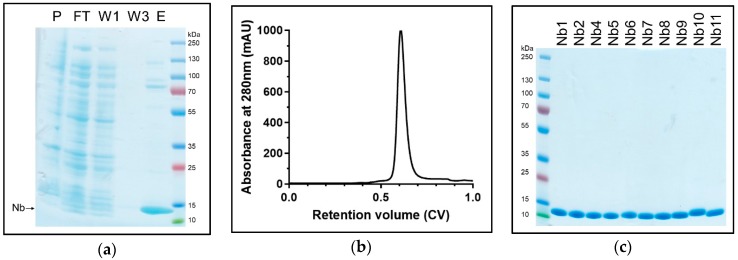
Nanobody purification. Samples from purification of Nb1 are representative of all the assessed Nbs. (**a**) Samples from insoluble fraction (pellet, P), flow-through of the IMAC (FT), first and third wash of Ni^2+^-beads (W1 and W3, respectively), and eluted fraction containing Nb1 loaded on Coomassie-stained SDS-PAGE. (**b**) SEC (24mL, Superose75) elution profile: The main peak elutes at 0.6 CV with only minor contaminants. (**c**) Coomassie-stained SDS-PAGE of SEC-purified Nbs: Nbs have a molecular weight of approximately 13.5 kDa.

**Figure 6 antibodies-07-00039-f006:**
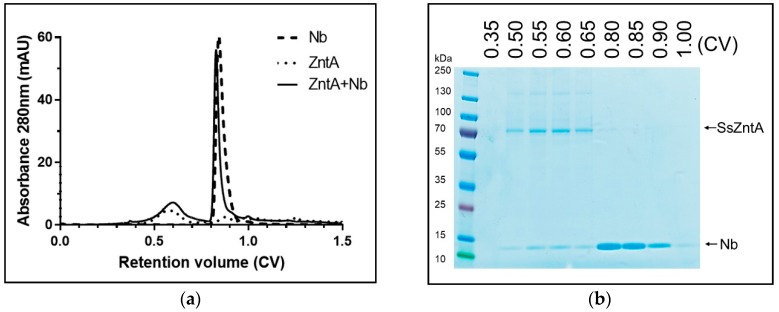
SsZntA-Nb complex analysis using size-exclusion chromatography. (**a**) Black line: Chromatogram of SsZntA (40 µM) incubated with Nb1 (100 µM) in a final volume of 50 µL (then diluted to 500 µM) and run through a size-exclusion column (Superose6, 1 CV = 24 mL) showing a peak at 0.6 CV, which contains the SsZntA-Nb complex, and a peak at 0.85 CV, which contains unbound Nb. The dashed line represents the chromatogram of Nb1 alone, while the dotted line the one from SsZntA alone (under the same condition). (**b**) Coomassie-stained SDS-PAGE gel of the fractions collected at 0.35, 0.5, 0.55, 0.6, 0.65, 0.8, 0.85, 0.9, 1 CV.

**Figure 7 antibodies-07-00039-f007:**
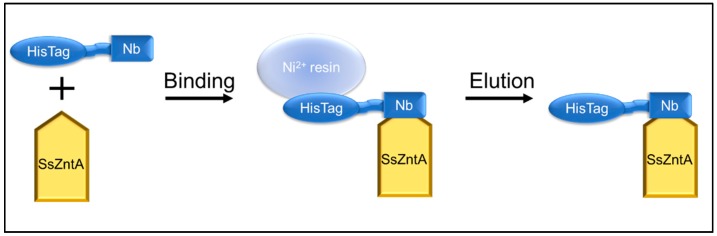
Affinity chromatography binding assay. SsZntA is treated with TEVp to remove the HisTag, in this way it cannot bind to a Ni^2+^ beads, while the HisTagged nanobody can. The binding was assessed by incubating HisTag-free SsZntA with HisTagged Nb and loading them to an Ni-NTA column. When SsZntA forms a complex with the Nb, it indirectly binds to the Ni^2+^ beads (via the Nb’s HisTag) and can be eluted from the Ni^2+^-immobilized resin. N.B. All the representations are out of scale for clarity.

**Figure 8 antibodies-07-00039-f008:**
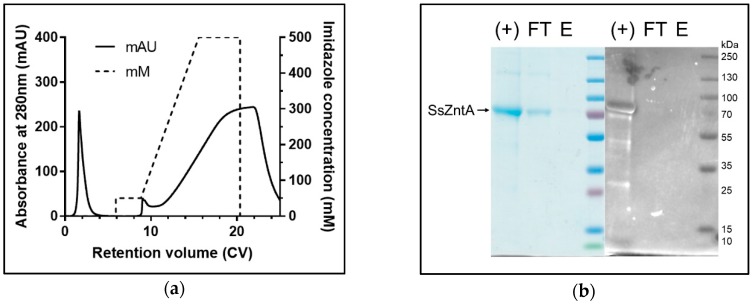
SsZntA-Nb complex analysis using affinity chromatography: SsZntA only. (**a**) Chromatogram of HisTag-free SsZntA loaded on an Ni^2+^-immobilized column. Without Nb, the ATPase elutes in the flow through (control). (**b**) Coomassie (left) and InVision^®^ (right) stained SDS-PAGE of HisTagged SsZntA as positive control for the staining (+), flow through (FT), and eluted fraction (E) of HisTag free SsZntA loaded on the Ni^2+^-immobilized resin as negative control.

**Figure 9 antibodies-07-00039-f009:**
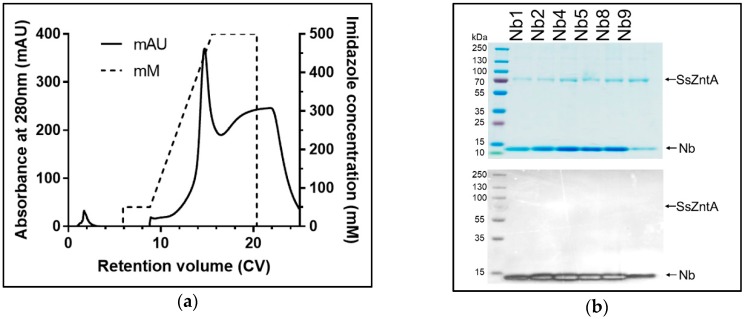
SsZntA-Nb complex analysis using affinity chromatography: SsZntA + Nb. (**a**) Chromatogram of HisTag-free SsZntA incubated with HisTagged Nb1. The complex elutes at 400 mM imidazole and is here reported as representative of the other Nbs. (**b**) Coomassie (top) and InVision^®^ (bottom) stained SDS-PAGE of the eluted fractions at 400 mM imidazole of HisTag-free SsZntA in complex with, respectively, Nb1-2-4-5-8-9.

**Figure 10 antibodies-07-00039-f010:**
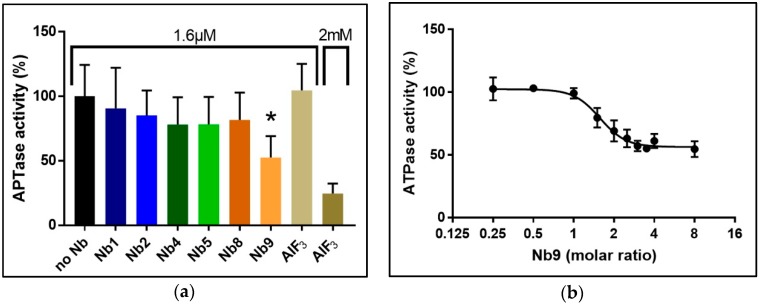
Nanobody effect on the catalytical function. (**a**) SsZntA was incubated with each Nb at a 1:2 molar ratio (i.e., 1.6 µM) and activity was assessed. Nb9 displayed an inhibitory effect of about 50% compared to untreated (no Nb) SsZntA. The difference between no Nb and Nb9 has a *p* value of <0.0001 (*), as assessed by a Dunn’s multiple comparison test. The ATPase inhibitor aluminum fluoride (AlF_3_) is shown as a positive control, at 1.6 µM (corresponding to the concentration of the tested Nbs) and at 2 mM. (**b**) The inhibition stoichiometry was assessed as titration of SsZntA with different molar ratios of Nb9. The inhibitory effect appears when Nb9 is present at a molar ratio of 1:1 (ZntA:Nb9), and reaches the inhibitory plateau at a molar ratio of 1:3 (ZntA:Nb9), with an IC_50_ = 1.57 molar ratio.

**Figure 11 antibodies-07-00039-f011:**
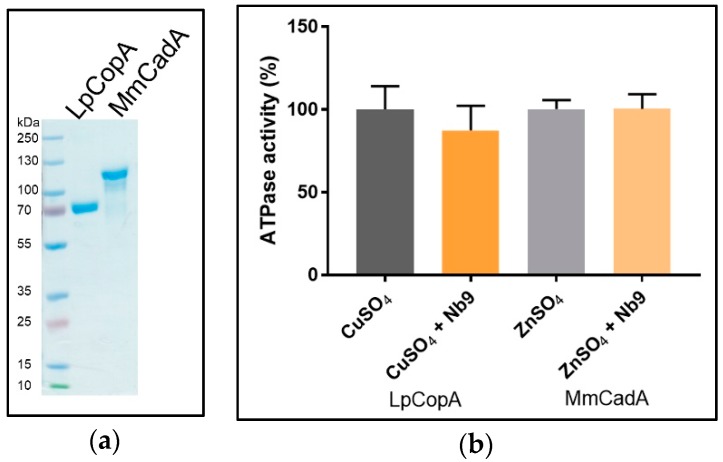
Nb9 specificity. (**a**) SDS-PAGE of purified Cu^+^-transporting P_IB-1_-ATPase from *Legionella pneumophila* (LpCopA) and Cd^2+^ and Zn^2+^-transporting P_IB-2_-ATPase from *Mesorhizobium metallidurans* (MmCadA) (**b**) Nb9 was tested for specificity against LpCopA and against MmCadA, in the presence of Cu^+^ or Zn^2+^, respectively. The results are normalized for the background in presence of the metal ion chelator EDTA.

**Table 1 antibodies-07-00039-t001:** Size-exclusion chromatography co-elution results.

Tested Nb	Binding	Precipitation
Nb1	+	-
Nb2	+	-
Nb4	+	-
Nb5	+	-
Nb6	N.D.	+
Nb7	N.D.	+
Nb8	+	-
Nb9	+	-
Nb10	N.D.	+
Nb11	N.D.	+

Not determined (N.D.): Due to sample precipitation, binding could not be assessed.

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
