# Peer review of "Isolation and Characterization of Nanobodies against a Zinc-Transporting P-Type ATPase"

_2073-4468, 2018, doi:10.3390/antib7040039_

Round 1

Reviewer 1 Report

This study was carried out to isolate nanobodies that selectively associate with SsZntA fromShigella sonnei.  SsZntA was cloned and expressed in E.coli.  The recombinant protein was purified and injected to llama. Nanobodies were screened and selected for binding and inhibiting SsZntA. It is a straight forward workflow and study. The methods were detailedly documented, and the results were well discussed. Overall, it is a complete story and it provides fundamental work for structural study in the future. However, there are some questions which I think integrity of this manuscript will be improved if they are properly addressed.

Here are my concerns.

Animals were involved in this study. Information of ethical approval should be provided.

Figure 10: positive control should be included in ATPase activity assay. ATPase inhibitor could be added, like vanadate or other specific inhibitors.

Figure 8 and Figure 9: they are not necessary, as previous results have already shown His-tag information on cleaved SsZntA and purified nanobodies.

SsZntA and Nb were incubated at the same moles (40uM each) in 2.4.1, but Figure 6 has shown there was excessive unbound Nb. How to explain this?

Figure 3: there are bands with higher molecular weight at 0.5CV and 0.6CV fractions. Are they dimer? How to explain this in SDS-PAGE gel?

For broader readers from various background to better understand the work, the difference between nanobodies and conventional antibodies should be in discussed in introduction.

Figure 2: the arrow of SsZntA doesn’t point right to the band.

Figure 2 and Figure 9: please leave a space between two images.

Please provide plasmid information for CopA and CadA.

Author Response

We thank the editor for the kind and swift consideration of our paper, and the three reviewers for recognizing the merit of our work and for providing very valuable critique. As detailed below, we have addressed their comments carefully and revised the manuscript accordingly. To facilitate the reviewing process, we have copied the Reviewers’ original comments, which are shown in black and our responses are then shown in red.

Please notice that we have changed one of the corresponding authors in the revised manuscript.  

This study was carried out to isolate nanobodies that selectively associate with SsZntA fromShigella sonnei.  SsZntA was cloned and expressed in E.coli.  The recombinant protein was purified and injected to llama. Nanobodies were screened and selected for binding and inhibiting SsZntA. It is a straight forward workflow and study. The methods were detailedly documented, and the results were well discussed. Overall, it is a complete story and it provides fundamental work for structural study in the future. However, there are some questions which I think integrity of this manuscript will be improved if they are properly addressed.

We thank the reviewer for the work and we are pleased to see that our work is recommended for publication.

However, there are some questions which I think integrity of this manuscript will be improved if they are properly addressed. Here are my concerns.

1.     Animals were involved in this study. Information of ethical approval should be provided.

We have revised the manuscript and included the following to clarify animal treatment procedures (page 3): “The immunization was performed by under the permit of Capralogics Inc. Capralogics Inc. provides a healthy housing environment for all animals and adheres strictly to USDA Animal Welfare Act regulations for Animal Care and Use.”

2.     Figure 10: positive control should be included in ATPase activity assay. ATPase inhibitor could be added, like vanadate or other specific inhibitors.

We thank the reviewer for this excellent suggestion. Figure 10 has now been revised and includes the effect of the known inhibitor aluminum fluoride (AlF3).

3.     Figure 8 and Figure 9: they are not necessary, as previous results have already shown His-tag information on cleaved SsZntA and purified nanobodies.

In our view, Figures 8 and 9 further substantiate the findings we show in Figure 6 and we therefore prefer to keep these data in the manuscript.

4.     SsZntA and Nb were incubated at the same moles (40uM each) in 2.4.1, but Figure 6 has shown there was excessive unbound Nb. How to explain this?

As reported in Figure 6, the SEC chromatogram is the result from an injection mixture containing 40 uM SsZntA and 100 uM Nb (molar ratio SsZntA:Nb of 1:2.5), and hence the excess unbound Nb is in line with the experimental design.

5.     Figure 3: there are bands with higher molecular weight at 0.5CV and 0.6CV fractions. Are they dimer? How to explain this in SDS-PAGE gel?

It is common to have higher oligomeric states (such as dimers) on SDS-PAGE of membrane proteins.

See for example https://www.ncbi.nlm.nih.gov/pmc/articles/PMC5715081/

6.     For broader readers from various background to better understand the work, the difference between nanobodies and conventional antibodies should be in discussed in introduction.

We thank the reviewer for this suggestion. A new paragraph (and references) has been included in the introduction section (page 2) to further explain the differences between nanobodies and classical antibodies.

7.     Figure 2: the arrow of SsZntA doesn’t point right to the band.

This request has now been addressed in the revised version of the manuscript.

8.     Please provide plasmid information for CopA and CadA.

The requested information has now been provided in the revised version of the manuscript.

Reviewer 2 Report

Authors successfully show Nb-induced inhibition of SsZntA.  This reviewer find couple minor comments to authors to further enhance the impact of the article-

Please include a control samples in SEC data presented in Fig. 6.  The controls shall be both the Nb and theSsZntA run individually under same SEC conditions.

2. Authors shall present more data on the inhibition kinetics (could be in form of isotherm).  Also, what is stoichiometry of the inhibition process?

Author Response

We thank the editor for the kind and swift consideration of our paper, and the three reviewers for recognizing the merit of our work and for providing very valuable critique. As detailed below, we have addressed their comments carefully and revised the manuscript accordingly. To facilitate the reviewing process, we have copied the Reviewers’ original comments, which are shown in black and our responses are then shown in red.

Please notice that we have changed one of the corresponding authors in the revised manuscript.  

Authors successfully show Nb-induced inhibition of SsZntA.  This reviewer find couple minor comments to authors to further enhance the impact of the article.

The positive recommendation from the reviewer is highly valued, and we have now revised the manuscript accordingly (see below).

1.     Please include a control samples in SEC data presented in Fig. 6.  The controls shall be both the Nb and theSsZntA run individually under same SEC conditions.

These controls have now been included in Figure 6.

2.     Authors shall present more data on the inhibition kinetics (could be in form of isotherm).  Also, what is stoichiometry of the inhibition process?

It will not be possible for us to provide isothermal titration calorimetry (ITC) measurements within the requested resubmission dead line of ten days – we do not regularly use the technique in our laboratory. We have attempted to respond to the inhibition stoichiometry question through titration of SsZntA with different molar ratios of Nb9 (applying the functional assay). The inhibitory effect appears when Nb9 is present at a molar ratio SsZntA:Nb9 of 1:1 and reaches an inhibitory plateau at a molar ratio SsZntA:Nb9 of 1:3, with an IC50 = 1.57, thus hinting at an inhibitory constant of 1 or 2.

Reviewer 3 Report

Manuscript titled Isolation and characterization of nanobodies against 
a zinc transporting P-type ATPase is a good attempt in generating nanobodies specific to  P-type ATPase. How ever this manuscript can be further improved by including points mentioned below.

1.In general binding affinity measurements were shown using at least one quantitative and one qualitative technique. Authors used only qualitative methods to show binding raises question. To strengthen the arguments of the manuscript authors should report binding affinity using routinely used SPR, Octet or ITC.

2.It is surprising to see nanobodies which do not bind also precipitated. Authors should explain what could be the possible explantation.

3.Inhibition of ATPase activity by NB9 could be due to higher binding affinity compare to other NBs. Can author compare binding affinity Vs inhibition ?

4.In Figure 8B  lane + in with InVision®
staining is showing a strong band suggests that presence of His-tag.  Does authors consider this band dis due to His tag ?

5. Experiments using LpCopA and MmCadA very interesting.

Author Response

We thank the editor for the kind and swift consideration of our paper, and the three reviewers for recognizing the merit of our work and for providing very valuable critique. As detailed below, we have addressed their comments carefully and revised the manuscript accordingly. To facilitate the reviewing process, we have copied the Reviewers’ original comments, which are shown in black and our responses are then shown in red.

Please notice that we have changed one of the corresponding authors in the revised manuscript.  

Manuscript titled Isolation and characterization of nanobodies against a zinc transporting P-type ATPase is a good attempt in generating nanobodies specific to  P-type ATPase. How ever this manuscript can be further improved by including points mentioned below.

We thank the reviewer for assessing our manuscript and considering our work relevant.

1.     In general binding affinity measurements were shown using at least one quantitative and one qualitative technique. Authors used only qualitative methods to show binding raises question. To strengthen the arguments of the manuscript authors should report binding affinity using routinely used SPR, Octet or ITC.

We address this question in the response to reviewer 2 (question 2) as follows: it will not be possible for us to provide isothermal titration calorimetry (ITC) measurements within the requested resubmission dead line of ten days – we do not regularly use the technique in our laboratory. We have attempted to respond to the inhibition stoichiometry question through titration of SsZntA with different molar ratios of Nb9 (applying the functional assay). The inhibitory effect appears when Nb9 is present at a molar ratio SsZntA:Nb9 of 1:1 and reaches an inhibitory plateau at a molar ratio SsZntA:Nb9 of 1:3, with an IC50 = 1.57, thus hinting at an inhibitory constant of 1 or 2.

2.     It is surprising to see nanobodies which do not bind also precipitated. Authors should explain what could be the possible explantation.

Nb3 precipitated already following affinity chromatography. Upon mixing of SsZntA with Nb6, 7, 10 or 11 we observed precipitation in the sample. After centrifugation, the sample was loaded on a SEC column and resulted in a “flat” chromatogram, indicating complete precipitation of Nb and SsZntA. Therefore, binding could not be assessed. We believe that the observed precipitation can relate to the presence of detergent, perhaps in combination of Nb binding which apparently compromises SsZntA further in this environment (e.g. through interaction with unstable intermediates or at sensitive regions of the proteins such as the membrane interface).

We have revised the manuscript to include a possible explanation in the discussion section (page 12 of the revised document).

3.     Inhibition of ATPase activity by NB9 could be due to higher binding affinity compare to other NBs. Can author compare binding affinity Vs inhibition ?

Yes, this is possible. We cannot exclude inhibition at higher Nb concentrations and we can in principle provide similar titration curves as for Nb9 (see Figure 10 b in the revised manuscript), but we cannot do this within the requested resubmission dead line of ten days. We included the following sentence at line 335: “The inhibition on the ATPase activity by the other Nbs is non-significant at the concentrations tested.” We also wish to note that since these Nbs are to be exploited for structural studies, SsZntA:Nb stoichiometries of 1:1 or 1:2 would be the most relevant.

4.     In Figure 8B  lane + in with InVision staining is showing a strong band suggests that presence of His-tag.  Does authors consider this band dis due to His tag ?

Yes, we do. Figure 8B, lane (+), includes a sample of IMAC-purified HisTagged SsZntA used as positive control for the InVision staining. The aim is to prove that this staining method allows detection of HisTagged SsZntA, and therefore the lack of signal in Figure 9B derives from the absence of HisTag in the co-eluted SsZntA. This has been better described in the section 3.4 of the results (affinity purification co-elution, page 9).

5.     Experiments using LpCopA and MmCadA very interesting.

We thank the reviewer for this positive remark.

Round 2

Reviewer 1 Report

In this revised manuscript, my questions have been addressed and additional information has been provided. It is an interesting study and now it is better organized. 

Thank you for your excellent job!

Reviewer 3 Report

Revised version looks much better. Thank you very making revisions in a quick time. 

One minor revision, Fig 10 A and B looks overlapping ( it may be the web version). Please double check.